# Robust deep learning-based gait event detection across various pathologies

**Bernhard Dumphart**[1,2,6]\*, **Djordje Slijepcevic**[3], **Matthias Zeppelzauer**[3], **Andreas Kranzl**[4], **Fabian Unglaube**[4], **Arnold Baca**[5], **Brian Horsak**[1,2]

**1** Center for Digital Health & Social Innovation, St. Pölten University of Applied Sciences, St. Pölten, Austria, **2** Institute of Health Sciences, St. Pölten University of Applied Sciences, St. Pölten, Austria, **3** Institute of Creative\Media/Technologies, St. Pölten University of Applied Sciences, St. Pölten, Austria, **4** Laboratory of Gait and Movement Analysis, Orthopaedic Hospital Vienna-Speising, Vienna, Austria, **5** Centre for Sport Science and University Sports, University of Vienna, Vienna, Austria, **6** Doctoral School of Pharmaceutical, Nutritional and Sport Sciences, University of Vienna, Vienna, Austria

\* bernhard.dumphart@fhstp.ac.at

## Abstract

The correct estimation of gait events is essential for the interpretation and calculation of 3D gait analysis (3DGA) data. Depending on the severity of the underlying pathology and the availability of force plates, gait events can be set either manually by trained clinicians or detected by automated event detection algorithms. The downside of manually estimated events is the tedious and time-intensive work which leads to subjective assessments. For automated event detection algorithms, the drawback is, that there is no standardized method available. Algorithms show varying robustness and accuracy on different pathologies and are often dependent on setup or pathology-specific thresholds. In this paper, we aim at closing this gap by introducing a novel deep learning-based gait event detection algorithm called *IntellEvent*, which shows to be accurate and robust across multiple pathologies. For this study, we utilized a retrospective clinical 3DGA dataset of 1211 patients with four different pathologies (malrotation deformities of the lower limbs, club foot, infantile cerebral palsy (ICP), and ICP with only drop foot characteristics) and 61 healthy controls. We propose a recurrent neural network architecture based on long-short term memory (LSTM) and trained it with 3D position and velocity information to predict initial contact (IC) and foot off (FO) events. We compared *IntellEvent* to a state-of-the-art heuristic approach and a machine learning method called DeepEvent. *IntellEvent* outperforms both methods and detects IC events on average within 5.4 ms and FO events within 11.3 ms with a detection rate of $\geq$ 99% and $\geq$ 95%, respectively. Our investigation on generalizability across laboratories suggests that models trained on data from a different laboratory need to be applied with care due to setup variations or differences in capturing frequencies.

## Introduction

An accurate and robust gait event detection algorithm is an important tool for minimizing extrinsic variability and time resources during post-processing of 3D gait analysis (3DGA) data. Gait cycle events are necessary for the calculation of spatiotemporal variables and a

**Data Availability Statement:** Raw trajectory data cannot be shared publicly because of restrictions due to hospital policy. Data requests may be sent to Dr. Andreas Kranzl (andreas.kranzl@oss.at)

from the Orthopaedic Hospital Vienna-Speising. The trained model and processing pipelines are available at: https://github.com/fhstp/IntellEvent.

**Funding:** This work received funding from the Austrian Gesellschaft für Forschungsförderung NÖ (Research Promotion Agency of Lower Austria) within the Endowed Professorship for Applied Biomechanics and Rehabilitation Research (SP19-004), the Science Call 2020 (SC20-020), and the Life Science Call 2018 (LSC18-018). The funders had no role in study design, data collection and analysis, decision to publish, or preparation of the manuscript.

**Competing interests:** The authors have declared that no competing interests exist.

proper segmentation and visualization of kinematic and kinetic variables. These variables are used by clinicians to evaluate and diagnose patients. The gait cycle consists of the stance and the swing phase, which are defined by two important events, namely the initial contact (IC) and the foot off (FO). The 'gold standard' for identifying these gait cycle events is by setting a predefined vertical ground reaction force (GRF) threshold on force plates. However, force plate contacts are sparse, compared to the whole capture space. Thus, often only a few measured steps can be utilized in this scenario. In many cases, force plate contacts are even unavailable due to pathological characteristics like foot drag, overly small steps, or the use of walking aids. In these cases, events are detected manually by trained clinicians through visual identification. However, depending on various factors, e.g., severity of pathology, capture space settings, and stride length, manual identification of gait events is prone to subjectivity and can be tedious and time-consuming work which can easily lead to human error [1].

To overcome these problems, various algorithms for automated event detection have been developed in the past. These approaches can be divided into heuristic and probabilistic approaches. Heuristic approaches rely on the occurrence of local minima and maxima in kinematic data (position, velocity, or acceleration) of markers of the lower extremities. A disadvantage of these algorithms is that a predefined threshold is required. Differences in walking speed, pathologies, and foot strike patterns affect marker trajectories to such an extent that the use of a single threshold for different patient groups may not be sufficient. Four individual studies [1–4] compared clinical heuristic event detection algorithms to identify the most robust and accurate method for a broad spectrum of pathological gait patterns. Out of seven different approaches [5–11], the modified version of Ghoussayni et al. [5] achieved the most promising results overall for both the FO and IC detection. However, results show varying robustness and accuracy for different pathological gait characteristics and in some cases exceed the commonly used tolerance window defined as a threshold of four frames [1–4]. Two further studies investigated the utility of heuristic approaches for self-paced treadmill walking [12] and straight-line walking with turning [13]. Both studies identified a combination of algorithms proposed by O'Connor et al. [8] and Zeni et al. [6] as the best approach for IC and FO detection. These findings further highlight the lack of consistency in the application of heuristic approaches for gait event detection. Due to the inconsistent performance across different pathologies, two studies [1, 3] suggested to first identify the most suitable algorithm for the underlying task and next to fine-tune it to the underlying pathology or to use two different approaches to detect the IC and FO events. The use of several different or optimized algorithms is associated with additional time and effort.

To overcome the issue of an input dependent predefined threshold, probabilistic-based approaches have been recently introduced to automatically detect gait cycle events. Approaches based on machine learning (ML) and deep learning methods, e.g., recurrent neural networks (RNNs), outperformed the most common heuristic approaches and showed more accurate and robust results. Lempereur et al. [2] investigated the performance of a stacked bi-directional long short-term memory (LSTM) neural network architecture and Kidziński et al. [14] utilized a stacked LSTM network to detect gait cycle events in children with mostly neurological disorders. Lempereur et al. [2] achieved mean accuracies of 5.5 ms and 10.7 ms for IC and FO events, respectively, compared to the best heuristic approach of Ghoussayni et al. [5] with 27.1 ms and 11.4 ms. Kidziński et al. [14] reported the mean accuracy of 10 ms and 13 ms for IC and FO events, respectively, outperforming the heuristic approach of Zeni et al. [6]. Filtjens et al. [15] utilized a temporal convolutional neural network to detect gait events during turning in patients with Parkinson's disease and freezing of gait and compared their method to the LSTM-based approach by Kidziński et al. [14] and the heuristic approach from Zeni et al. [6]. Kim et al. [16] introduced a bi-directional LSTM approach for

different striking patterns (with the heel, midfoot, and toe) in 363 children with infantile cerebral palsy (ICP) with an overall detection rate for IC and FO events of 89.7% and 71.6%, respectively, within ± 16ms of the ground truth. Furthermore, Kim et al. [16] evaluated different marker setups. For certain ICP subgroups, specific marker setups yielded up to a 5–10% increase in performance. The results of the aforementioned studies also confirm that RNN models are more accurate and robust compared to the heuristic approaches.

The above-mentioned ML-based approaches, however, have not been evaluated on different patient groups separately, offering no insight into how well these algorithms perform on different pathologies. The problem in this regard is that if the algorithm under investigation performs poorly on a rather small group of patients, this would not be visible in the averaged results reported in the literature. Since clinical datasets are often imbalanced in terms of the size of different patient groups, such an analysis is necessary for daily use in clinical practice. To date, such analyses have only been conducted for heuristic approaches [1, 3]. To fully understand the capabilities and limitations of automated gait event detection methods, it is important to investigate accuracy and robustness across different patient groups (e.g., pathologies) in the underlying dataset.

To close the above-mentioned gaps we present a novel gait event detection algorithm that we call "*IntellEvent*", which is based on two bi-directional LSTM models trained separately for each gait event type. The proposed algorithm is evaluated on a variety of different pathological gait patterns. The first goal of this work is to compare our proposed event detection algorithm to state-of-the-art methods, i.e., the heuristic algorithm proposed by Ghoussayni et al. [5] and the ML-based approach *DeepEvent* from Lempereur et al. [2] on different gait pathologies including malrotation deformities of the lower limbs, club foot, infantile cerebral palsy (ICP), ICP with only drop foot characteristics, as well as healthy controls.

In addition to evaluating the accuracy and robustness of the proposed method, we investigate the generalizability of state-of-the-art ML methods with respect to different gait laboratories. Minor differences in the general setup or marker placement may introduce a bias learned by the event detection algorithm, affecting the accuracy of the algorithm when applied to data from different laboratories. Visscher et al. [17] showed this for heuristic approaches, where a difference of up to 15 ms occurred between laboratories. Smaller differences in results were shown in laboratories with a more comparable setup [17]. Therefore, it is important to understand how a model performs between different gait laboratories. On the one hand for the general usability of a method but on the other hand for validity and reproducibility purposes as well.

Hence, the second goal of this work is to evaluate the state-of-the-art ML approach DeepEvent [2] on our own dataset (i.e., without retraining the model). For direct comparison, the same model architecture is also retrained and evaluated on our data.

Our investigation focuses on two main research questions:

- RQ1: Can selected ML algorithms detect gait events in different pathologies robustly and with comparable performance levels that are at least equivalent to state-of-the-art results?

- RQ2: How well can selected ML-based algorithms detect gait events on data from a different laboratory on which they were not trained?

The main contribution of this work can be summarized as follows: (i) We evaluate the performance of ML-based gait event detection methods on various pathologies using one of the most comprehensive datasets in the literature, (ii) we propose a novel event detection approach that is based on two separate deep learning models for the two event types that are trained using only ground truth events from force plates as compared to state-of-the-art

methods, which have used all available (e.g., manually set) events, and (iii) we explore the presence of bias in a publicly available state-of-the-art ML model utilizing data from our gait laboratory to understand current limitations of ML-based gait event detection methods.

The source code for *IntellEvent* is available on GitHub (https://github.com/fhstp/IntellEvent) with detailed documentation for training, retraining, and integrating the algorithm in an existing motion capturing pipeline. Thereby, we aim at offering a simple, flexible, and easy-to-use event detection approach as a baseline for further research and implementation into various existing gait analysis routines with established processing workflows and clinical standards.

## Methods

### Data and preprocessing

This study utilised a retrospective clinical 3DGA dataset of 1211 patients and 61 healthy controls (male: 664, female: 608, age: 18 ± 14.6 years). All persons were examined by a clinician and categorized into one of five classes depending on the underlying pathology—malrotation deformities of the lower limbs (MD, n = 730), club foot (CF, n = 120), infantile cerebral palsy (ICP n = 344), ICP with only drop foot characteristics (DF, n = 17), and healthy controls (HC, n = 61). Overall, the dataset contains 5717 trials (with different numbers of trials per person) with at least four gait events per trial determined by force plates. A trial is defined as one recording including several consecutive steps in one direction. Only measurements from the first visit of a patient and trials without shoes and any kind of walking aides were utilized (cp. Table 1). 3D gait data was recorded on a 12-meter walkway at self-selected walking speed using 12 infrared cameras (Vicon Motion Systems Ltd, Oxford Metrics, UK) and three strain-gauge force plates (Advanced Mechanical Technology Inc., Watertown, MA) with a sampling rate of 150 Hz and 1500 Hz, respectively. For capturing kinematic data of the lower extremities, the extended Cleveland Clinic marker set [18] was applied in combination with the Vicon Plug-In-Gait model for the upper body. The ground force detection threshold was set to 20 N. 3D trajectories of the markers were filtered using the Woltring filtering routine integrated in the Vicon Nexus system with an MSE value of 15.

This retrospective study used anonymized clinical gait data from an existing database maintained by a local Austrian hospital. Prior to the experiments involved and the publication of the results, approval was obtained from the local Ethics Committee of the city of Vienna (EK19–083-VK).

### Event detection with neural networks

Our gait event detection algorithm *IntellEvent* is based on a stacked bi-directional LSTM architecture (cp. Fig 1). For the detection of IC and FO events, two separate models were trained. The reason for this was, that when using only one model trained on the ground truth force

**Table 1. Descriptive information of the persons in the dataset.** Weight, height, and age are reported as mean values (± standard deviation).

| | Malrotation Deformities | Club Foot | Drop Foot | ICP | Healthy Controls |
|---|---|---|---|---|---|
| **Amount** | 730 | 120 | 17 | 344 | 61 |
| **Trials** | 3205 | 461 | 66 | 1454 | 531 |
| **Weight [kg]** | 48.6 (20.2) | 40.8 (21.2) | 42.0 (25.6) | 44.5 (20.1) | 63.5 (24.9) |
| **Height [mm]** | 1511.8 (213.9) | 1376.8 (275.6) | 1420.6 (216.8) | 1448.4 (244.6) | 1563.2 (430.0) |
| **Age [years]** | 12.4 (7.2) | 12.0 (9.7) | 9.5 (14.1) | 14.1 (9.9) | 20.2 (6.9) |

**Fig 1. Network architecture of *IntellEvent*.** Two models with such architecture were trained for the two different event types (IC and FO). The input dimension is 577x36, whereas 577 is the length of the longest input trial in the training data, and 36 is the number of input channels, i.e. the trajectories (heel, toe, ankle) and their derivatives. The output sequence for the model are three nodes. Each node outputs the probability of either no gait event, a left IC/FO event, or a right IC/FO event. Gait events are detected from the output probabilities with a simple peak detection using a threshold of 0.01.

plate data to detect all events, problems occurred in detecting the first FO and last IC events in most trials. The use of two separate models mitigates this problem and provides further benefits by allowing different hyperparameters and different model architectures to be used for the two different events.

To find a suitable configuration for our architecture, hyperparameters of the stacked LSTM model were optimized by conducting a grid search including the number of layers (i.e., 1, 2, 3), hidden units (i.e., 50, 100, 150, 200, 250, 300, 350), dropout rates (i.e., 0.2, 0.4, 0.6), and sample weights (ratio of non-events to events: 1:10, 1:100, 1:500, 1:1000). For the weighted cross-entropy loss, sampling weights were used to compensate for the large imbalance in the data, since the number of non-events (i.e., all frames without a specific gait event) was very high compared to the number of relevant events (ratio of an event to a non-event is 1:191). Results from the grid search yielded the optimized configuration of three bi-directional LSTM layers with 200 hidden units, each followed by a dropout layer (dropout rate of 0.4), and sample weights ratio of 1:10. Finally, a time-distributed dense layer with softmax as an activation function feeds three output nodes. Each node outputs the probability of either no gait event, a left IC/FO event, or a right IC/FO event. Model parameters were optimized to minimize the categorical cross-entropy loss using the Adam optimizer. For performance reasons, a masking layer was added which ignores zeros in the input data stream created by zero-padding and thereby significantly speeds up processing. A visualization of the architecture used is shown in Fig 1. Gait events were detected from the output probabilities with a simple peak detection algorithm using the same threshold (0.01) as proposed by Lempereur et al. [2]. The following list contains the general observations we made for the individual hyperparameters that we evaluated during the grid search:

- **Number of layers**: A single-layered model performed worst for IC and FO. Two- and three-layered models performed similarly for IC, but three-layered models were more consistent. Three-layered models performed best for FO.

- **Number of hidden units**: The IC model performed best with 200 and 300 hidden units. The FO model performed best with 200 hidden units.

- **Dropout rate**: The FC model showed similar performances with different dropout rates, but a dropout rate of 0.4 showed more consistent results. The FO model showed similar performances with dropout rates of 0.2 and 0.4.

- **Sample weights**: The IC model performed best with a weight ratio of 1:10, while higher ratios resulted in a drop in performance. The FO model performed similar with a weight ratio of 1:10 and 1:100 and performed worse with higher ratios.

From the extended Cleveland Clinical marker set [18], the 3D trajectories of the left and right heel, toe, and ankle markers as well as their first derivatives were used as inputs for the RNN leading to 36 channels overall. Each training and validation trial was cropped to contain only events detected by the force plate with a random frame buffer between 25 and 125 frames before and after the first and last event. We investigated the effect of the buffer size on the temporal error and detection rate. These limits were determined empirically, as the inclusion of buffers below 25 and above 125 frames had no positive effect on performance. To prepare the data for the training of the neural network, data were reshaped using zero-padding (right-hand side) to match the longest trial. Furthermore, we normalized each channel to a range between 0.1 and 1.1.

For the training of the neural network, the whole dataset was split into a training (60%), a validation (10%) and a test (30%) set. Data were stratified at the patient level to ensure that an individual's data was only contained in either the training, validation, or test set. Due to the overall low amount of DF data, we decided to use this class only in the test set as an out of sample class to evaluate the generalization ability of the detector in a zero-shot setting.

*IntellEvent* was implemented within the software framework Python 3.8.5 (Python Software Foundation, USA) and TensorFlow 2.3.0 (Google Inc., USA) using a workstation running Ubuntu 20.04.3 LTS with one GeForce GTX 1080 (8 GB RAM), with 128 GB of system memory and an Intel(R) Core(TM) i7–6900K CPU @ 3.20GHz.

## Implementation of state-of-the-art methods

To compare *IntellEvent* with a heuristic method, we implemented the modified approach of Ghoussayni et al. [5]. This approach uses a predefined threshold for heel and toe marker velocity to detect IC and FO events. To determine the optimal thresholds that depend on walking speed, a grid search was performed on the training data. The determined thresholds (TH), i.e., $TH_{IC} = 0$ and $TH_{FO} = 1.9v$ (with $v$ being the walking speed), were used on the test data, allowing for a fair comparison between all methods.

For the comparison of our approach with a previously proposed ML-based approach, DeepEvent [2] was implemented in two different ways. Firstly, to ensure a direct comparison, DeepEvent was implemented according to the original model architecture and further trained from scratch on our training data and evaluated on our test data (see Section Data and preprocessing). We refer to this model in the following as *DeepEvent-retrained*. Note that in contrast to our architecture, DeepEvent-retrained was trained on all available gait events in each trial (of the training set) leading to almost three times as much training data with the longest trial consisting of 1525 frames. This training approach is consistent with the original definition of Lempereur et al. [2]. Hence, trials contained not only ground truth data from the force plates but also manually identified gait events or those determined by autocorrelation, which were subsequently inspected manually.

Secondly, we investigate how well the original (pretrained) DeepEvent model without additional adjustments performs on data from our gait laboratory and on different gait pathologies. We refer to this model in the following as *DeepEvent-original*. For this comparison, the pretrained model was acquired from the corresponding author and the original code for preprocessing was downloaded from GitHub (https://github.com/LempereurMat/deepevent). Subsequently, the model was then evaluated on our adequately preprocessed test data.

## Data analysis

For the comparison of the different approaches, the test dataset comprised only gait events detected by the force plates. This ground truth information was compared to the events

detected by *IntellEvent*, DeepEvent-original and DeepEvent-retrained, as well as the approach of Ghoussayni et al. [5]. To investigate how well a certain model can detect gait events, we utilized two performance metrics. Namely the temporal error and the detection rate. The temporal error, i.e., the difference between the ground truth and the predicted events, was calculated using the mean absolute error (MAE) and is specified in milliseconds (ms). According to the capturing frequency of 150 Hz, 6.7 ms corresponds to one frame. Furthermore, we reported the lower and upper limits of the 95% confidence interval (CI). The detection rate is defined as the number of correct predictions (true positives, TPs) divided by the total number of predictions, i.e., the sum of all TPs, false positives (FPs), and false negatives (FNs). A TP was counted if an event was predicted within the four-frame tolerance window (i.e., ±26.7 ms) around the corresponding ground truth event. An FP was counted if an event was predicted within a 50-frame window (i.e., ±333.3 ms) around the corresponding ground truth event. If the two thresholds defined above were not met, an FN was counted.

After identifying the best combination of hyperparameters from the extensive grid search, a sensitivity analysis of the best performing network was conducted. The optimal network was trained and tested ten times with a different random seed to identify the impact of random initialization on the current setup. A random seed is a number used to initialize the pseudo-random number generator, generating different random starting points (i.e., initialization) for the ML model. This allows an evaluation of the robustness of the neural network with respect to different initializations. To this end, with a standard deviation between 0.14 and 1.06 ms, we can conclude that our network performs robustly. The results of this analysis can be found in the (S1 Appendix) section. For visualization and presentation purposes, the best performing model observed in our investigation will be presented in Section Results.

### Statistical analysis

All data relevant for statistical testing were substantially not normally distributed, indicated by a Shapiro-Wilk test and visual inspection. Therefore non-parametric tests were used. To evaluate if there is a statistically significant difference in IC and FO detection between the *IntellEvent*, DeepEvent-retrained, and the approach of Ghoussayni et al. [5] we used pair-wise Wilcoxon signed rank tests. This was conducted separately for IC and FO events and for each pathology. All reported *p*-values were Bonferroni corrected by a factor of three to mitigate alpha inflation. To test if there is a statistically significant difference in the performance between DeepEvent-original and DeepEvent-retrained a pair-wise Wilcoxon signed rank test was used, separately for IC and FO events and each condition. Additionally, the effect size was calculated using the Pearson correlation coefficient *r* as defined by Cohen [19].

## Results

Results for all event detection approaches in terms of temporal error and detection rate are presented in Figs 2 and 3. Overall, 8022 events (4011 ICs and 4011 FOs) were analyzed. The class distributions of the ground truth events are as follows: malrotation deformities of the lower limbs: 2237, club foot: 329, infantile cerebral palsy: 987, ICP with only drop foot characteristics: 139, and healthy controls: 319.

### Temporal error

Both *IntellEvent* and DeepEvent-retrained showed low MAEs compared to the ground truth across all pathologies for IC and FO event detection (cp. Fig 2); The IC event was detected on average within 2.5 to 6.6 ms and FO between 7.9 to 11.9 ms (cp. Tables 2 and 3 ). In comparison to *IntellEvent* and DeepEvent-retrained, the DeepEvent-original model showed generally

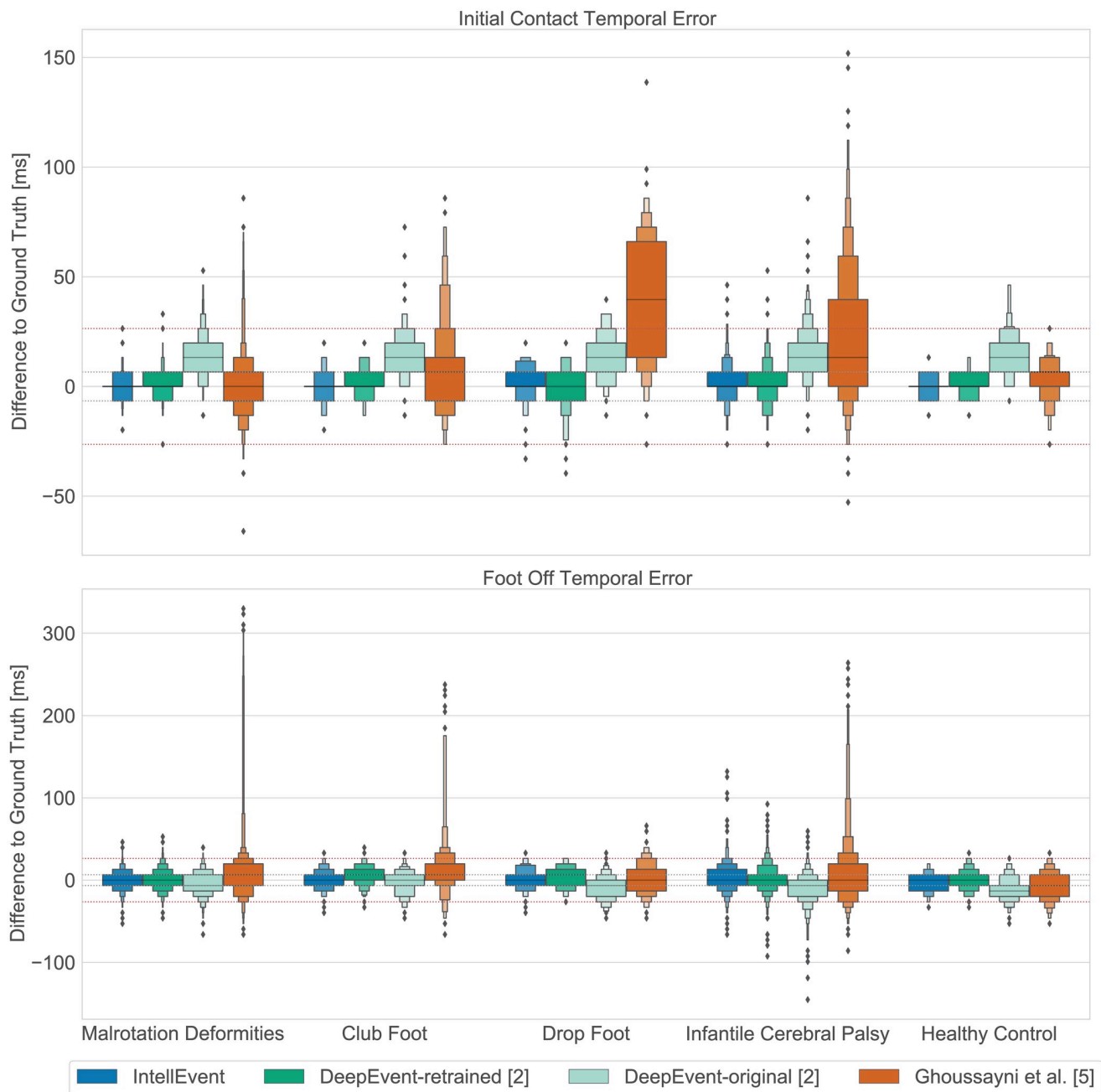

**Fig 2. Temporal error, as the difference of the predicted event to the force plate ground truth in milliseconds, for each approach and pathology separately, for initial contact event detection (top) and foot off event detection (bottom).** Our method (*IntellEvent*) is displayed in blue (first), the DeepEvent-retrained model [2] in green (second), the DeepEvent-original model [2] in light green (third), and the method of Ghoussayni et al. [5] in orange (forth). The horizontal dotted line (red) indicates the tolerance window of 26.7 ms.

higher MAEs with a temporal error of 13.1 to 16.1 ms for IC and 10.5 to 15.6 ms for FO events. The heuristic approach of Ghoussayni et al. [5] showed the highest MAE for almost all pathologies for the IC (7.3 to 41.5 ms) and FO events (15.0 to 23.3 ms) compared to the other methods. In nine out of 4011 cases, *IntellEvent* classified IC events as FP with the worst event being 46.7 ms off compared to the ground truth. Eight events belong to the ICP class, and one event

**Fig 3. Cumulative detection rate for each approach and pathology shown separately, for the initial contact event detection (top) and the foot off event detection (bottom).** Our method (*IntellEvent*) is displayed in blue (first), the DeepEvent-retrained model [2] in green (second), the DeepEvent-original model [2] in light green (third), and the method of Ghoussayni et al. [5] in orange (forth). The x-axis defines the tolerance window in frames (0: 0 ms, 0–1: 0 ms—6.7 ms, 0–2: 0 ms—13.3 ms, 0–3: 0 ms—20.0 ms, 0–4: 0 ms—26.7 ms) and the y-axis the cumulative detection rate in percent.

**Table 2. Mean absolute error of the predicted initial contact (IC) events compared to the ground for each method and pathology.** All values are reported in milliseconds (95% CI). Bold values represent the lowest mean absolute error in each pathology.

|  | Malrotation Deformities | Club Foot | Drop Foot | ICP | Healthy Control |
|---|---|---|---|---|---|
| *IntellEvent* | **2.7 (2.5–2.8)** | **3.5 (3.0–3.9)** | **5.4 (4.4–6.4)** | **4.9 (4.6–5.3)** | **2.5 (2.1–2.9)** |
| **DeepEvent-retrained** | 3.1 (3.0–3.3) | 3.9 (3.4–4.4) | 6.6 (5.4–7.9) | 5.9 (5.5–6.2) | 3.7 (3.3–4.2) |
| **DeepEvent-original** | 14.0 (13.7–14.3) | 13.1 (12.1–14.1) | 16.1 (14.5–17.8) | 16.0 (15.4–16.7) | 14.9 (13.8–15.9) |
| **Ghoussyani et al. [5]** | 8.8 (8.4–9.2) | 14.0 (12.3–15.8) | 41.5 (36.9–46.1) | 27.0 (25.4–28.7) | 7.3 (6.6–7.9) |

**Table 3. Mean absolute error of the predicted foot off (FO) events compared to the ground truth for each method and pathology.** All values are reported in milliseconds (95% CI). Bold values represent the lowest mean absolute error in each pathology.

|  | Malrotation Deformities | Club Foot | Drop Foot | ICP | Healthy Control |
|---|---|---|---|---|---|
| *IntellEvent* | **7.9 (7.6–8.2)** | **8.7 (7.9–9.5)** | **9.9 (8.5–11.3)** | **11.3 (10.5–12.1)** | **8.3 (7.6–9.1)** |
| **DeepEvent-retrained** | 9.0 (8.7–9.4) | 9.3 (8.4–10.2) | 10.1 (8.8–11.4) | 11.9 (11.1–12.6) | 8.9 (8.1–9.7) |
| **DeepEvent-original** | 11.4 (11.0–11.8) | 10.5 (9.5–11.5) | 13.7 (11.9–15.6) | 14.8 (13.9–15.7) | 15.6 (14.5–16.8) |
| **Ghoussyani et al. [5]** | 19.0 (17.7–20.3) | 20.2 (16.9–23.6) | 17.2 (15.0–19.4) | 23.3 (21.2–25.4) | 15.0 (13.7–16.3) |

belongs to the DF class. For FO events, 99 out of 4011 events were classified as FP where most of the false positive events belong to the ICP class (54 events) followed by the MD class (28 events). For the FO events that were classified as FP, the average error is 43.5 ms, with 9 events over 66.7 ms and the worst detection being 133.3 ms off.

### Detection rate

Results for the detection rate are summarized in Fig 3. *IntellEvent*, as well as DeepEvent-retrained demonstrated IC and FO detection rates above 97% and 93%, respectively. Similar to the temporal error, the detection rate of DeepEvent-original for IC and FO events was lower, ranging from 88% to 96% with no pathology reaching 100%. The heuristic approach of Ghoussayni et al. [5] presented the lowest detection rate, which also highly varied for the different pathologies, ranging from 36% to 100% for IC and FO events.

### Statistical analysis

A Wilcoxon signed rank test indicated a statistically significant difference ($p < .001$) in the IC events between *IntellEvent* and DeepEvent-retrained for all classes except for CF and DF. The FO was only statistically significantly different ($p < .001$) in the LLM class. However, there is a generally low effect size for this comparison ($r = .01$—.31). Similarly, when comparing *IntellEvent* and DeepEvent-retrained with the heuristic algorithm, the Wilcoxon signed rank test indicated significant differences ($p < .001$) for all classes, but with higher effect sizes ($r = .20$ —.82).

Between DeepEvent-original and DeepEvent-retrained models, a Wilcoxon signed rank test indicated statistically significant differences for the IC prediction ($Z < -7.33$, $p < .001$) and high effect sizes ($r > .61$) for all classes. Furthermore, on average a difference of 10.2 ms was observed between the two models for the IC detection (cp. Fig 2). For the FO predictions, four classes, namely LLM, DF, ICP, and HC, indicated significant differences ($Z < -7.0$, $p \leq .001$) and small to high effect sizes ($r > .22$), whereas CF indicated no significant differences ($p > .086$) and low effect size ($r < .09$). All the corresponding results can be found in the (S1 Appendix).

## Discussion

In this study, we introduced and evaluated *IntellEvent* a novel gait event detection method based on recurrent neural networks. The first goal was to evaluate our model together with several state-of-the-art alternatives [2, 5] and compare their performance on a dataset comprising various pathologies (RQ1). For deep learning-based approaches, this is the first study that directly compares different approaches across multiple pathologies. The second goal was to investigate the generalizability of a state-of-the-art deep learning-based approach [2] using data recorded in different gait laboratories (RQ2).

### Event detection across pathologies

For the usefulness in daily clinical practice, it is important to validate the performance of gait event detection algorithms independent of pathology. With our findings we can confirm, that deep learning-based methods are capable of detecting IC and FO events in various pathologies with a low temporal error and a high level of detection rate. Moreover, our results show that deep learning-based methods perform significantly better than heuristic approaches, which is also consistent with previous comparisons between these two groups of approaches [2, 14], highlighting the advantages of such methods and their robustness.

Similarly to the findings of Lempereur et al. [2] and Kim et al. [16], our results show that the FO detection was consistently less accurate than the IC detection. In this regard, Kidzinski et al. [14] pointed out that the subjectivity and quality of manually set ground truth events could influence FO detection. In our case, only ground truth events determined by force plates were utilized, which should have eliminated any subjectivity factor. Fig 3 shows for each investigated method the cumulative detection rate over the individual tolerance windows. For the best-performing method, *IntellEvent* IC events are detected with 54% within 0 ms on average, whereas FO events are detected only with 23% on average. Furthermore, FO events mostly need a tolerance window of up to 3 frames to achieve a detection rate of 90%, whereas IC events can be predicted with 90% using a tolerance window of one frame only (except for DF and IPC, which require 2 frames). This comparison indicates that the toe, ankle, and heel marker alone could be insufficient for FO detection. Visscher et al. [20] analyzed different marker placements on healthy persons and patients, to identify the best combination for IC and FO detection. The results show, that the use of midfoot markers reduces IC and FO detection errors compared to the classical heel and toe marker placement. Especially in patients who use the side of the forefoot or the midfoot (equinus) for IC, the error can be reduced by 50% [20]. In combination with ML approaches, the use of midfoot markers could result in even more accurate and robust event detection algorithms for different pathologies. However, up to this date, most of the conventional markersets in clinical practice are neither using midfoot nor hallux markers [20]. With further development (and standardization) of the conventional gait model and its marker set 2.0 [21], this problem could be mitigated and has to be analyzed in future.

Compared to the other classes, *IntellEvent* exhibits the lowest overall performance in patients with ICP, as evidenced by the highest temporal error and the lowest detection rate. Nevertheless, in terms of lower temporal errors for IC and FO detection (IC: 4.9 ms, FO: 11.3 ms), these results are better compared to the results of Goncalves et al. [4] (IC: 13.49 ms, FO: 17.3 ms). The ML-based approach of Kim et al. [16] showed comparable results, but there model was trained only on ICP patients. The lower detection rate of the FO events in the ICP class could be due to suboptimal marker placement as explained above. ICP patients mostly are forefoot/midfoot IC strikers, which could have negatively influenced the results due to no midfoot or hallux marker as described by Visscher et al. [20]. Kim et al. [16] tested different marker placements for cerebral palsy patients with an LSTM-based approach and reported a 5–10% increase in accuracy when using subgroup-specific foot markers (hallux and/or midfoot).

In addition, the higher temporal error of the ICP class in *IntellEvent* could be explained by the low weight of patients in this group and the 20 N threshold of the force plates. 70% of the ICP patients in our dataset are 14 years old or younger and the average mass of this patient group is 36 (± 14.2) kg. The ground truth of very light patients may have been influenced by a threshold that corresponds to 5.7% of the average body weight for this patient group. During the pre-swing (shortly before the foot leaves the ground) there are lower forces compared to the foot strike. Thus, the foot of ICP patients could have been still on the ground but the force could have already been below 20 N resulting in an offset. False negative results where this potentially influenced the ground truth were checked manually by the clinicians and afterwards corrected. Still, all other trials where the FO detection is below the false negative threshold could have inherited the same bias and therefore could have influenced the temporal error of the ICP class. For future comparison, force plate thresholds for very light people should be adapted based on their body weight. Visscher et al. [1] analyzed the impact of different force plate thresholds on gait event detection where errors of up to 4.9 ms for both events can occur, depending on the pathology. Furthermore, a high agreement between different force

thresholds was reported ($R^2$ = .99) but no information about the body weights of the participants was provided.

A side-result concerning the DF class, showed that *IntellEvent* has the potential to identify gait cycle events on a pathology that has not been present during method training. As this class comprises data from only 17 patients, the results need to be considered with caution. Nevertheless, the analysis of this class is interesting because DF patients have a distinct foot strike pattern (forefoot) that differs from other pathologies (rearfoot/midfoot) in our dataset.

With a detection rate of 97% and a temporal error of 5.4 ms and 9.9 ms for the IC and FO events, respectively, the performance level of *IntellEvent* on this unseen pathology is similar to that of the ICP class which the model was trained on. Furthermore, these results are comparable to the findings of Kim et al. [16] who actively trained their model on a DF class. This underlines the strong generalization ability of *IntellEvent* to an unseen pathology. The approach by Ghoussayni et al. [5] (cp. Figs 2 and 3) yields considerably lower performance and cannot compete with deep learning-based approaches in terms of generalization ability to an unseen pathology.

Regarding RQ1, we conclude that deep learning-based methods (in particular recurrent architectures) are superior to heuristic approaches and can identify gait events with a lower temporal error and a higher detection rate across different pathologies.

### Generalizability across laboratories

A particular challenge for event detection algorithms seems to be their generalization ability across different gait laboratory settings. RQ2 directly addresses this topic. Our results indicate that generalization is also a problem for deep learning-based approaches. Fig 2 shows that DeepEvent-retrained outperformed DeepEvent-original when evaluated on our dataset (cp. Tables 2 and 3). This difference is significant for IC events ($p < .001$, $r > .61$) for all classes and for FO events ($p < .001$, small to high effect sizes) for all classes except for CF. A more detailed comparison of the cumulative detection rate in Fig 3 shows for IC detection that DeepEvent-original has a very low detection rate in lower tolerance windows (0: 8%, 0–1: 33%) compared to the DeepEvent-retrained model (0: 45%, 0–1: 88%). The retraining on the target data improves performance, which shows that the network can successfully adapt to the new data. At the same time, this shows that the original model cannot generalize well to the data from another laboratory. One reason for the poorer performance could be that the data from the two laboratories exhibit two different distributions due to the difference in recording frequency. The data from Lempereur et al. [2] was captured at 100 Hz, whereas our data was captured at 150 Hz. This highlights the importance of future approaches being robust to different recording frequencies.

Furthermore, the marker placement and setup differences between laboratories as well as the manually set gait cycle events by professional clinicians could also have added a certain bias as reported in [17]. To date, only the heuristic approach of Ghoussayni et al. [5] was compared between nine different gait laboratories [17]. With an absolute error of 10 ms for IC and 17.5 ms for FO estimation between different gait laboratories this bias could also translate to deep learning-based approaches as it is comparable to the difference of 10.2 ms for the IC between the DeepEvent-original and DeepEvent-retrained models in our results. Laboratories with similar setups and marker placement protocols show higher agreement in the results [17]. Therefore, Visscher et al. [17] suggested the need for a multi-center approach to improve reproducibility and validity.

Since DeepEvent does not utilize any normalization of trajectories [2], this could result in the inability to account for different laboratory sizes, different setups, and slight differences in

marker placement. To mitigate these biases, we incorporated normalization strategies into the *IntellEvent* approach.

With respect to RQ2, we conclude that using available deep learning-based approaches, which have been trained on data from a specific laboratory, need to be applied to different gait laboratories with care because a certain bias could be introduced due to different sampling frequencies or setup differences between the laboratories. Although we have introduced some mechanisms in *IntellEvent* to minimize these influencing factors (trajectory normalization, training only on ground truth data from force plates), *IntellEvent* could also be subject to these biases. It is therefore essential to develop novel mitigation strategies for gait event detection algorithms as well as validation routines that can objectively assess differences in the distributions of data from different laboratories.

## Ground truth verification

Aside from the high detection rate and low temporal errors of *IntellEvent* on a majority of the data, results also show a few sparse outliers where performance breaks, see Figs 2 and 3. We manually checked these cases to identify the reason for the low performance. In our experiments, *IntellEvent* detected 36 outliers for the IC events and 192 outliers for the FO events out of 8022 events (4011 IC, 4011 FO). Trials with detected outliers were then manually checked for errors in the ground truth. Using this method, we were able to correct 11 IC and 70 FO events in the dataset, where the events were incorrectly set on the GRF plate due to reasons like foot drag, force plate thresholding, or force plate artefacts.

The verification of the ground truth was significantly accelerated by using the detection results from *IntellEvent* and would have not been feasible at this level otherwise. Thus, data verification and outlier analysis is an additional interesting application of automated gait event detection algorithms.

## Limitations and future work

For FO detection the marker placement of the toe marker seems to be the limiting factor in achieving a more accurate and robust event detection. With our approach, we tried to mitigate subjectivity and ensure high quality [14] of the FO events by using only ground truth events determined from the force plates. Still, FO detection shows a lower accuracy and robustness compared to IC detection. With the use of a midfoot or a hallux marker, a more accurate FO detection has been reported [16, 20], but hallux and midfoot markers have not been standardized and are not available in most current clinical marker sets. Additionally, DF and ICP patients show the highest temporal error for IC detection over all pathologies. With ICP patients showing a diverse range of gait patterns (true equinus, apparent equinus, crouch gait and jump knee) leading to different foot strike patterns. Thus, a midfoot marker could also further improve IC detection [20]. Future work in this context should investigate the effect of different marker combinations as well as different sets of input features for both the IC and FO model. Furthermore, as *IntellEvent* consists of two separate models for the two gait events, future work will include the investigation of combined model architectures and the joint optimization of both detection tasks. Since *IntellEvent* outperforms all evaluated state-of-the-art approaches, we provide it as open source (https://github.com/fhstp/IntellEvent) for the community to be used as a new performance baseline.

A limiting factor in our study is the low quantity of DF patients. For this reason, the *IntellEvent* was not trained on DF data. These data were only used in the test dataset. Nevertheless, our approach showed promising results for a pathological class which was not used in the training process. To assess the robustness of *IntellEvent* for daily clinical practice, future work

should focus on a systematic evaluation of the generalization ability to previously unseen pathologies.

As demonstrated in our experiment, there is a need for a multi-center approach to gait event detection that is suitable for different gait laboratory settings. Future work should investigate the influence of differences in the general setup, marker placements, and capturing frequencies. ML-based gait event detection is most promising to work robustly in different gait laboratories regardless of these influencing factors. Two different approaches could be explored, either creating one algorithm that is optimized on different gait laboratory data during training or using transfer learning to adapt an algorithm from one laboratory to another. For transfer learning an interesting question is how much labeled training data is necessary to adapt a detector to a new laboratory and further if unsupervised pre-training helps to improve the generalization ability across laboratories.

## Conclusion

In this study, we introduce a novel deep learning-based gait event detection method called *IntellEvent* that is based on LSTMs for the automated detection of initial contact and foot off events. We evaluated the performance of *IntellEvent* across different pathologies and compared its performance to the state-of-the-art heuristic approaches of Ghoussayni et al. [5] and the recently introduced DeepEvent of Lempereur et al. [2]. Furthermore, we evaluated how well a ML-based method can perform on data from a laboratory on which it was not trained. *IntellEvent* showed highly accurate results for IC event detection within 5.4 ms and for FO events within 11.3 ms on average with a detection rate of $\geq$ 99% and $\geq$ 95%, respectively. Based on our results, we can conclude that selected deep learning-based approaches outperform heuristic approaches for gait event detection and show promising results in terms of their ability to generalize to unseen pathologies. Finally, our evaluation indicated that the performance of deep learning-based approaches depends on laboratory setups, which opens up space for further research.

## Supporting information

**S1 Appendix. This file contains supporting descriptive and statistical tables.**
(ZIP)

## Author Contributions

**Conceptualization:** Bernhard Dumphart, Djordje Slijepcevic, Matthias Zeppelzauer, Andreas Kranzl, Fabian Unglaube, Arnold Baca, Brian Horsak.

**Data curation:** Bernhard Dumphart, Djordje Slijepcevic, Andreas Kranzl, Fabian Unglaube.

**Formal analysis:** Bernhard Dumphart, Djordje Slijepcevic, Matthias Zeppelzauer, Brian Horsak.

**Funding acquisition:** Bernhard Dumphart, Brian Horsak.

**Methodology:** Bernhard Dumphart, Djordje Slijepcevic, Matthias Zeppelzauer, Andreas Kranzl, Fabian Unglaube, Arnold Baca, Brian Horsak.

**Project administration:** Brian Horsak.

**Resources:** Andreas Kranzl, Brian Horsak.

**Supervision:** Matthias Zeppelzauer, Arnold Baca, Brian Horsak.

**Validation:** Bernhard Dumphart.

**Visualization:** Bernhard Dumphart.

**Writing – original draft:** Bernhard Dumphart.

**Writing – review & editing:** Bernhard Dumphart, Djordje Slijepcevic, Matthias Zeppelzauer, Andreas Kranzl, Fabian Unglaube, Arnold Baca, Brian Horsak.

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
