## [Decision Letter · Decision Letter 0]

17 Feb 2023

PONE-D-22-35192Robust deep learning-based gait event detection across various pathologiesPLOS ONE

Dear Dr. Dumphart,

Thank you for submitting your manuscript to PLOS ONE. After careful consideration, we feel that it has merit but does not fully meet PLOS ONE’s publication criteria as it currently stands. Therefore, we invite you to submit a revised version of the manuscript that addresses the points raised during the review process. Please revise and resubmit your manuscript.

We look forward to receiving your revised manuscript.

Kind regards,

Kathiravan Srinivasan

Academic Editor

PLOS ONE

Journal Requirements:

2. Please ensure that you have specified (1) whether consent was informed and (2) what type you obtained (for instance, written or verbal, and if verbal, how it was documented and witnessed). If your study included minors, state whether you obtained consent from parents or guardians. If the need for consent was waived by the ethics committee, please include this information.

5. We note that the manuscript is reporting a meta-analysis on genetic association studies. We need you to provide us with additional information in relation to this meta-analysis; please complete the following checklist and upload it as a Supporting Information file with a file name “GAMA checklist”. The checklist can be downloaded here: http://www.plos.org/wp-content/uploads/2013/05/meta-analysis-on-genetic-association-studies-form.docx

Reviewers' comments:

Reviewer's Responses to Questions

**Comments to the Author**

1. Is the manuscript technically sound, and do the data support the conclusions?

Reviewer #1: Yes

Reviewer #2: Yes

Reviewer #3: No

2. Has the statistical analysis been performed appropriately and rigorously? 

Reviewer #1: Yes

Reviewer #2: Yes

Reviewer #3: Yes

3. Have the authors made all data underlying the findings in their manuscript fully available?

Reviewer #1: No

Reviewer #2: Yes

Reviewer #3: No

4. Is the manuscript presented in an intelligible fashion and written in standard English?

Reviewer #1: Yes

Reviewer #2: Yes

Reviewer #3: No

5. Review Comments to the Author

Reviewer #1: This paper proposes a deep learning approach for gait analysis, and more specifically for the detection of events in gait signals acquired in the laboratory. The experiments are conducted on a large (inaccessible) clinical dataset and a link is provided to the codes used (however, this link does not seem to be working currently). The proposed approach is compared to two state-of-the-art methods and allows to evaluate the interest of deep learning approaches for this application. Moreover, it is shown that the learning transfer problem is an open issue for this application. The authors searched for the best architecture using a grid search method. It would have been interesting to have an idea of the performance of the studied architectures, and in particular to understand the impact of the architecture design on the performance. Is this a crucial point to obtain reliable performances in clinical routine?

Have the authors addressed the problem of uncertainty estimation in this context, to further quantify the possibility of introducing these learning approaches into clinical routine? And for example, what are the worst detections? Are they related to a particular walking profile? Are the errors similar?

Reviewer #2: The authors have developed a new deep learning (DL)-based algorithm (IntellEvent) to robustly identify gait cycle events from an optical motion capture system using reflective markers. The authors have performed a very detailed analysis of the algorithm performance in detecting events (foot-off, FO, and initial contact, IC) across different pathological gait patterns and compared it to both a validated heuristics-based algorithm and a relevant DL-based algorithm (i.e., DeepEvent). Besides testing its validity, the authors also address the applicability of a DL-based algorithm on an out-of-distribution dataset by evaluating the performance of DeepEvent on a new gait dataset.

Their newly proposed DL-algoritm (IntellEvent) achieved excellent results, and as such the authors have developed a sound and useful algorithm for events detection across different pathologies.

To further improve the manuscript, the authors may consider addressing the following point that were raised:

Introduction

Lines 49 – 52

The authors state that the modified version of Ghoussayni et al. achieved the most promising results overall for both the FO and IC detection.

They may want to refer that other research (Ulrich et al., 2019, doi: 10.1016/j.jbiomech.2019.05.006, Hendershot et al., 2016, J. Biomech. 49, 4146-4149) has shown that a combination of (O’Connor et al., 2007, Gait & Posture) for IC detection and (Zeni et al., 2008, Gait & Posture) for FO detection works best for gait events detection during treadmill walking and overground turning.

This actually strengthens the line of argumentation from the authors, that there is no consensus on which heuristics-based algorithm works best for event detection from marker trajectories.

Methods

Lines 149 – 152

The authors state that two separate models were trained, because otherwise problems occurred in detecting the first FO and the last IC in most trials. Could this be because the initial and terminating step are different from the “intermediate” or “steady-state” steps? In some gait research they leave out the initial and final two to three steps of analysis. Would this have been an option as well for the case that only one model is used?

Discussion

Lines 427 – 430

The authors state that “ … With respect to RQ2, we conclude that using available deep learning-basedapproaches, which have been trained on data from a specific laboratory, need to be applied to different gait laboratories with care because a certain bias could be introduced due to different sampling frequencies or setup differences between the laboratories.”

Does this mean that using IntellEvent on our gait dataset will also be prone to bias, in other words the use of IntellEvent is also limited to a certain marker and walking test setup?

Reviewer #3: In the paper the authors proposed a novel deep learning-based gait event detection algorithm called IntellEvent based on stacked bi-directional LSTM.

However, I am unable to find any novelty in this work.

(1) In line number 66-67, the authors mentioned that "Kidzinski et al. [12] utilized a stacked bi-directional long short-term memory (LSTM) neural network architecture to detect gait cycle events in children with mostly neurological disorders". Then, how there proposed stacked bi-directional LSTM is a novel approach.

(2) In line number 12-123, the authors mentioned "The source code for IntellEvent is available on GitHub (https://github.com/fhstp/IntellEvent) with detailed documentation for training, retraining, and integrating the algorithm in an existing motion capturing pipeline". However, the link is giving error 404 message.

(3) I line number 156-158, the authors mentioned To find a suitable configuration for our architecture, hyperparameters of the stacked LSTM model were optimized by conducting a grid search including the number of layers, hidden units, dropout values, and sample weights". However, the authors did not mention the best hyperparameter values.

(4) Explanation of figure 1 is totally missing.

(5) How the proposed stacked bi-directional LSTM architecture is different from the architecture that was proposed in Kidzinski et al. [12].

(5) The tables and figures should be in the proper position of the manuscript.

6. PLOS authors have the option to publish the peer review history of their article (what does this mean?). If published, this will include your full peer review and any attached files.

Reviewer #1: No

Reviewer #2: No

Reviewer #3: **Yes: **Dipanwita Thakur

---

## [Author Response · Author response to Decision Letter 0]

3 May 2023

Dear Reviewers,

please find attached the response to reviewers´ comments in the corresponding file (Response to Reviewers.pdf).

Yours sincerely,

Bernhard Dumphart

---

## [Decision Letter · Decision Letter 1]

29 Jun 2023

Robust deep learning-based gait event detection across various pathologies

PONE-D-22-35192R1

Dear Dr. Dumphart,

We’re pleased to inform you that your manuscript has been judged scientifically suitable for publication and will be formally accepted for publication once it meets all outstanding technical requirements.

Kind regards,

Kathiravan Srinivasan

Academic Editor

PLOS ONE

Additional Editor Comments (optional):

Reviewers' comments:

Reviewer's Responses to Questions

**Comments to the Author**

1. If the authors have adequately addressed your comments raised in a previous round of review and you feel that this manuscript is now acceptable for publication, you may indicate that here to bypass the “Comments to the Author” section, enter your conflict of interest statement in the “Confidential to Editor” section, and submit your "Accept" recommendation.

Reviewer #2: All comments have been addressed

Reviewer #4: (No Response)

Reviewer #5: (No Response)

2. Is the manuscript technically sound, and do the data support the conclusions?

Reviewer #2: Yes

Reviewer #4: Partly

Reviewer #5: Yes

3. Has the statistical analysis been performed appropriately and rigorously? 

Reviewer #2: Yes

Reviewer #4: No

Reviewer #5: Yes

4. Have the authors made all data underlying the findings in their manuscript fully available?

Reviewer #2: No

Reviewer #4: No

Reviewer #5: Yes

5. Is the manuscript presented in an intelligible fashion and written in standard English?

Reviewer #2: Yes

Reviewer #4: Yes

Reviewer #5: Yes

6. Review Comments to the Author

Reviewer #2: Congratulations to the authors on the excellent research manuscript. They have stressed an important gap in literature, namely that any (deep learning) algorithm for marker-based gait event detection suffers from bias. The authors have well addressed all the comments.

Reviewer #4: 1. It is claimed that there are no standardized methods for event detection. For the events mentioned, there are standard algorithms such as the Peak Detection Algorithm, and ZUPT (Zero update velocity). Apply these, compare, and justify your novelty.

2. References are not state-of-the-art after 2021. Work-related to the above algorithms should be part of this research.

3. Computation/analysis related to 3D velocity and position found missing, taken from the dataset. This should be supported by mathematical calculations applied to the dataset.

Reviewer #5: The goal of the paper was to predict the gait event (initial contact and foot off) in 3D gait analysis data. The authors used a deep learning approach to perform their task. The LSTM network was prepared and its hyper-parameters were tuned using the grid search approach. Experimental results were given along with comparison to a state-of-the-art solution and distinction to several pathologies.

In my opinion the manuscript focus on interesting problem and Authors completed their task correctly. However, I would like to point to some issues that, in my opinion, could improve the paper.

1) In the section "Data analysis" confusion matrix elements were specified. However, those definitions are not exactly clear. False Positive and False Negative conditions should be verified and corrected. If not, some strong elaboration should be included about why those conditions were specified in such a way.

2) I think it will be beneficial to include those results (TP, FP, FN) in some tabular way. This will allow a better comparison with the state-of-the-art models.

3) In line 221 the Authors stated that they separated validation subset from their data. However, no results were given from the validation experiments.

Additionally, in my opinion, the results given in the Appendix about different random seeds are redundant. Well-prepared deep learning models should give similar results despite the random seed. Instead of this, it would be beneficial to perform cross-validation (k-fold, for example). This will truly allow one to assess the robustness of the model. Since this would probably require a lot of additional work, I leave this as an optional suggestion.

Therefore, I recommend to accept the paper after revision.

7. PLOS authors have the option to publish the peer review history of their article (what does this mean?). If published, this will include your full peer review and any attached files.

Reviewer #2: No

Reviewer #4: No

Reviewer #5: No

---

## [Editor Report · Acceptance letter]

3 Aug 2023

PONE-D-22-35192R1 

Robust deep learning-based gait event detection across various pathologies 

Dear Dr. Dumphart:

I'm pleased to inform you that your manuscript has been deemed suitable for publication in PLOS ONE. Congratulations! Your manuscript is now with our production department. 

Kind regards, 

on behalf of

Dr. Kathiravan Srinivasan 

Academic Editor

PLOS ONE